# Alpha Satellite RNA Levels Are Upregulated in the Blood of Patients with Metastatic Castration-Resistant Prostate Cancer

**DOI:** 10.3390/genes13020383

**Published:** 2022-02-20

**Authors:** Sven Ljubić, Antonio Sermek, Angela Prgomet Sečan, Marin Prpić, Blanka Jakšić, Jure Murgić, Ana Fröbe, Đurđica Ugarković, Isidoro Feliciello

**Affiliations:** 1Department of Molecular Biology, Ruđer Bošković Institute, Bijenička 54, HR-10000 Zagreb, Croatia; sven.ljubic@irb.hr (S.L.); antonio.sermek@irb.hr (A.S.); ugarkov@irb.hr (Đ.U.); 2Department of Oncology and Nuclear Medicine, University Clinical Hospital Centre (UHC) Sestre Milosrdnice, Vinogradska cesta 29, HR-10000 Zagreb, Croatia; angela.prgomet.secan@kbcsm.hr (A.P.S.); marin.prpic@kbcsm.hr (M.P.); blanka.jaksic@kbcsm.hr (B.J.); jure.murgic@kbcsm.hr (J.M.); 3School of Dental Medicine, University of Zagreb, Gundulićeva 5, HR-10000 Zagreb, Croatia; 4Dipartimento di Medicina Clinica e Chirurgia, Universita’ degli Studi di Napoli Federico II, Via Pansini 5, 80131 Napoli, Italy

**Keywords:** alpha satellite DNA, transcription, alpha satellite RNA, prostate cancer, blood biomarker

## Abstract

The aberrant overexpression of alpha satellite DNA is characteristic of many human cancers including prostate cancer; however, it is not known whether the change in the alpha satellite RNA amount occurs in the peripheral tissues of cancer patients, such as blood. Here, we analyse the level of intracellular alpha satellite RNA in the whole blood of cancer prostate patients at different stages of disease and compare it with the levels found in healthy controls. Our results reveal a significantly increased level of intracellular alpha satellite RNA in the blood of metastatic cancers patients, particularly those with metastatic castration-resistant prostate cancer relative to controls. In the blood of patients with localised tumour, no significant change relative to the controls was detected. Our results show a link between prostate cancer pathogenesis and blood intracellular alpha satellite RNA levels. We discuss the possible mechanism which could lead to the increased level of blood intracellular alpha satellite RNA at a specific metastatic stage of prostate cancer. Additionally, we analyse the clinically accepted prostate cancer biomarker PSA in all samples and discuss the possibility that alpha satellite RNA can serve as a novel prostate cancer diagnostic blood biomarker.

## 1. Introduction

Satellite DNAs are tandemly repeated sequences predominantly located within constitutive heterochromatin which is positioned in the (peri)centromeric and subtelomeric regions of chromosomes. Alpha satellite DNA is the major human satellite DNA that makes up 3–5% of each human chromosome and is composed of basic units based on divergent, 171 bp-long monomers [1]. Alpha satellite DNA contributes to essential chromosomal functions such as the formation of the centromere/kinetochore as well as of constitutive heterochromatin [2]. The heterochromatin structure is defined by epigenetic modifications, in particular by H3K9me3, which is catalysed by the histone methyltransferase SUV39H1 [3]. Of particular importance is the role of alpha satellite DNA transcripts whose interaction with the enzyme SUV39H1 is necessary for the proper formation and regulation of heterochromatin not only in the pericentromeric regions [4], but also at alpha satellite repeats dispersed within euchromatin [5].

Numerous diseases including cancers can result from deregulated epigenetic mechanisms, which also affect the structure of pericentromeric heterochromatin and the expression of sequences located therein. For instance, the lysine-specific demethylase 2A (KDM2A)—which is specific for H3K36—is a tumour suppressor and is downregulated in prostate cancer [6]. The lower the level of *KDM2A* expression, the higher pericentromeric heterochromatin transcription and the more severe the tumour grade in prostate cancer [6]. The aberrant overexpression of sequences within pericentromeric heterochromatin in which the satellite DNAs including alpha satellite predominate is not only characteristic of prostate cancer but also of many other epithelial cancers such as those of the pancreas, lung, kidney, and colon [7]. The expression of satellite DNA in tumours can also occur due to a deficiency of other tumour suppressors such as p53 or BRCA1 which disrupts the integrity of constitutive heterochromatin and results in an extremely high expression of satellite DNAs [8,9]. Increased levels of satellite RNA destabilise the replication fork and genome integrity and further promote tumour transformation [10]. It should be mentioned that the overexpression of human satellite DNAs is not only characteristic of tumours but has also been observed in replicative senescence and aging correlating with the loss of H3K9me3 at the pericentromeric heterochromatin [11,12]. In addition, the strong upregulation of satellite DNA expression occurs upon heat stress [5,13,14,15,16].

Although the aberrant overexpression of alpha satellite DNA in many human cancers including prostate cancer has been described, it is not known whether changes in alpha satellite RNA amounts occur in the peripheral tissues of cancer patients, such as blood, and whether they could be used as potential cancer biomarkers. For the diagnosis of prostate cancer, the most clinically accepted biomarker is prostate-specific antigen (PSA). PSA is a kallikrein-related serine protease produced by prostate epithelial cells, whose levels are usually elevated in prostate cancer patients. The introduction of PSA testing in asymptomatic men has resulted in the earlier detection of the disease, with a reduction in the percentage of men with metastatic prostate cancer. PSA also allows for the early detection of latent prostate cancer that often does not develop into a significant disease and is often elevated under benign conditions such as inflammation or hyperplasia, and this lack of PSA specificity results in over-diagnosis [17]. PSA is also used to monitor the development of the disease, although often its level, especially in patients with metastatic prostate cancer, neither closely correlates with the stage of the disease nor with hormone-sensitivity. Due to the insufficient specificity of PSA as a diagnostic and prognostic marker, additional efforts are being made to find alternative biomarkers for prostate cancer [18].

To gain insight into the possible use of alpha satellite RNA as a prostate cancer biomarker, we analysed the level of intracellular alpha satellite RNA in the blood cells of cancer prostate patients and compared it with the levels of healthy controls. Our results reveal that the level of alpha satellite RNA is significantly upregulated in the blood of patients with metastatic, castration-resistant prostate cancer relative to healthy controls, indicating a link between prostate cancer pathogenesis and intracellular alpha satellite RNA levels in blood. In the blood of patients with metastatic hormone-sensitive prostate cancer, an increase in alpha satellite RNA levels was detected but not statistically significant. Testing at other stages of disease revealed no significant change of alpha satellite levels relative to controls. The clinically accepted prostate cancer biomarker PSA was also determined in all blood samples and obtained values compared to those of alpha satellite RNA levels. We discuss a possible mechanism which could be related to the increased level of intracellular alpha satellite RNA in the blood of patients at a specific stage of prostate cancer as well as the possibility that alpha satellite RNA can serve as an indicator of a particular pathophysiological state or as a potential novel prostate cancer diagnostic blood biomarker.

## 2. Materials and Methods

### 2.1. Sample Collection

The blood samples of prostate cancer patients were provided and collected by Prof. Dr. Sc. Ana Fröbe from the University Clinical Hospital Centre (UHC) Sestre Milosrdnice. The blood samples of healthy controls were provided by the Croatian Institute for Transfusion Medicine (HZTM). Informed consent was acquired from each participating individual before blood collection. Ethical approval was obtained from the Medical Ethical Committees of the UHC Sestre Milosrdnice and of the Croatian Institute for Transfusion Medicine.

We collected 2.5 mL of blood from each cancer prostate patient and healthy control in a “PAXgene Blood RNA Tube” (Qiagen) according to the instructions of the manufacturer which was stored at −20 °C. This study comprised a total of 94 patients with prostate cancer diagnosis and 27 healthy controls. Since 50% of men older than 50 years have benign prostate hyperplasia [19], which can result in an increased PSA level, we used men with an average age of 39.4 years as the healthy control group. Enrolled patients belong to different groups according to the stage of disease: A—metastatic hormone-sensitive; B—metastatic castration-resistant; and C—localised hormone-sensitive. All patients in groups A–C received androgen-deprivation therapy by luteinising hormone-releasing hormone (LHRH) agonists. Group D included patients with newly diagnosed localised prostate cancer before receiving a hormone or any local treatment.

Control subjects were male blood donors with no personal history of prostate cancer or any other chronic disease and at the time of blood collection—they were not on a drug therapy. Other patients’ and controls’ characteristics that were documented included age. The characteristics of the patients and controls are presented at Table 1.

### 2.2. RNA Isolation and Reverse Transcription

Intracellular RNA from whole blood collected in the PAXgene Blood RNA Tubes was isolated using the PAXgene Blood RNA Kit (Qiagen, Hilden, Germany) according to the instructions of the manufacturer. RNA was quantified with the Quant-IT RNA assay kit using a Qubit fluorometer (Invitrogen, Waltham, MA, USA)). Approximately 1 μg of RNA was reverse transcribed using the PrimeScript RT reagent Kit with gDNA Eraser (Takara, Dalian, China) in 10 μL reaction using the specific primer ALPHrev AATGCACACATCACAAAGAAG. For all samples, negative controls without reverse transcriptase were used.

### 2.3. Quantitative Real-Time PCR (qPCR) Analysis

qPCR analysis was performed according to the previously published protocol [5,15]. Primers for the expression analysis of human alpha satellite DNA were: ALPHAfw CACTCTTTTTGTAGAATCTGC and ALPHrev AATGCACACATCACAAAGAAG, which were constructed according to the alpha satellite consensus sequence [20]. *Glucuronidase β (GUSB*) [21] was used as an endogenous control for normalisation in human samples and the primers used were: GUSfw GAAAATATGTGGTTGGAGAGAGCATT, GUSrev CCGAGTGAAGATCCCCTTTTTA. *GUSB* gene (Gene ID: 2990) was stably expressed without any variation among samples. The following thermal cycling conditions were used: 50 °C 2 min; 95 °C 7 min; 95 °C 15 s; 60 °C 1 min for 40 cycles followed by dissociation stage: 95 °C for 15 s; 60 °C for 1 min; 95 °C for 15 s; and 60 °C for 15 s. Amplification specificity was confirmed by dissociation curve analysis and the specificity of amplified products was tested on agarose gel. Control without template (NTC) was included in each run. Post-run data were analysed using LinRegPCR software v.11.1. [22,23] which enables the calculation of the starting concentration of amplicon (“no value”). No value is expressed in arbitrary fluorescence units and is calculated by considering PCR efficiency and baseline fluorescence. “No value” determined for each technical replicate was averaged and the averaged “no values” were divided by the “no values” of the endogenous control. The statistical analysis of qPCR data was performed using GraphPad v.6.01.

### 2.4. Determination of PSA Values in Blood

Peripheral blood samples were collected by the venepuncture of cubital vein in the sitting position punctured by one laboratory staff person, in 6 mL Vaccuete^®^ serum tubes with clot activator (red cap), (Greiner Bio-One, Kremsmünster, Austria) according to national recommendations for venous blood sampling. Serum samples were centrifuged within 4 h and the analyses of PSA were made within 2 h of centrifugation. Blood for serum testing was centrifuged for 10 min at 2150× *g* at 4 °C on Hettich ROTINA35 centrifuge (Hettich, Germany). Sera samples were measured on Roche Cobas e601 (Roche Diagnostics GmbH, Mannheim, Germany) automated immunochemistry analyser with analytical principle of electrochemiluminescence reaction, using original Roche assays, calibrators, and controls.

### 2.5. Statistical Analyses

The Shapiro–Wilk test was used to test data normality. Statistical differences in PSA values as well as in the alpha RNA level in four groups of patients and controls were tested using Kruskal–Wallis test. The alpha satellite RNA level of patients belonging to four groups and of the control group were displayed in boxplots and were tested for statistical significance using the parametric 2-tailed Welch’s *t* test if the data had normal distribution (controls, groups A and B) and non-parametric Mann–Whitney test if the data had non-normal distribution (groups C and D). The correlation between the alpha satellite RNA and PSA levels was assessed using Spearman’s rank correlation. The diagnostic potential of the alpha satellite RNA level for distinguishing metastatic prostate cancer patients and controls were evaluated by computing receiver operating characteristic (ROC) curves and the results were quantified by the area under the curve (AUC) in the pROC [24] package. The statistical analyses were performed using R software [25] and graphs were created using ggplot2 [26] package. *p*-values less than 0.05 were considered statistically significant.

## 3. Results

### 3.1. Alpha Satellite RNAs Level in the Blood of Prostate Cancer Patients—qPCR Analysis

We isolated intracellular RNA from whole blood which was collected from prostate cancer patients belonging to four groups representing different stages of disease: A—representing patients with metastatic hormone-sensitive prostate cancer; B—patients with metastatic castration-resistant cancer; C—patients with localised hormone-sensitive prostate cancer; and D which included patients with newly diagnosed localised prostate cancer before receiving hormone or any other treatment. In contrast to group D, patients from groups A–C were all under hormone treatment (LHRH agonists). In parallel, we isolated intracellular RNA from the control groups which included 27 healthy male individuals. The number of samples of each group, average age as well as age range is shown in Table 1.

To measure the level of alpha satellite RNA in the total intracellular RNA isolated from the whole blood of patients from four different groups (A–D) as well as from healthy controls, we used quantitative real-time PCR (qPCR) analysis. The obtained qPCR results (Appendix A) were analysed by Kruskal–Wallis test which is used to analyse the differences among multiple groups of samples: control and four groups of patients, and it revealed significant difference (*p* = 1.4 × 10^−^^4^). The 2-tailed Welch’s *t* test and Mann–Whitney test were used to see the difference among the pairs of samples and its statistical significance. The results reveal the increased level of alpha satellite RNA in two groups of metastatic prostate cancer patients relative to the control group (Figure 1). An increase of 2.8× with significant statistical support (*p* = 2.7 × 10^−^^4^) is characteristic of group B—with metastatic castration-resistant prostate cancer—while for group A—with metastatic hormone-sensitive prostate cancer—the increase is 1.4× but not statistically significant (*p* = 0.11). Within the other two groups of localised prostate cancers, namely C and D, there is no statistically significant difference in the level of alpha satellite RNA relative to the control (*p* > 0.05). The similarity in the alpha satellite RNA levels in groups C and D suggest that the level is not affected by drug treatment. There is also a significant increase in alpha satellite RNA level in group B relative to group A—metastatic hormone-sensitive prostate cancer of 2.0× (*p* = 4 × 10^−3^) as well as to groups C and D of 2.9× (*p* = 4 × 10^−6^) and 1.7× (*p* = 0.017), respectively.

The results show that the level of alpha satellite RNA can be used to distinguish between different stages of disease: metastatic castration-resistant relative to the metastatic castration-sensitive as well as metastatic castration-resistant relative to localised prostate cancer and to the healthy controls, respectively, and could serve as a potential diagnostic biomarker of metastatic state, particularly of castration-resistant metastatic prostate cancer. ROC curves and the calculation of AUC values (Figure 2) reveals that based on alpha satellite RNA levels, metastatic castration-resistant prostate cancer can be discriminated with high accuracy from primary localised tumours (AUC 0.85) and controls (AUC 0.85). Discrimination between metastatic castration-resistant and metastatic hormone-sensitive prostate cancer (B vs. A) is also acceptable (AUC 0.74).

### 3.2. PSA Values in Four Groups of Prostate Cancer Patients

The most clinically accepted biomarker for prostate cancer is prostate-specific antigen (PSA) which is a kallikrein-related serine protease produced by prostate epithelial cells. PSA levels are usually elevated in prostate cancer patients. We checked PSA values in the blood of four groups of patients (A–D) as well as controls and performed statistical analyses of log PSA values by Kruskal–Wallis test which is used to analyse differences among multiple groups of samples: control and four groups of patients, and it revealed significant difference (*p* = 1.7 × 10^−^^11^). The Mann–Whitney test revealed no significant difference between the PSA level in group B relative to group A (*p* = 0.9074) and to group D (*p* = 0.5063), while the PSA level in the three groups A, B, and D is significantly increased relative to the control and to group C (*p* < 10^−^^4^). PSA values between control and group C are not significantly different (*p* = 0.300; Figure 3).

The ROC curve analysis of PSA levels revealed discrimination between the controls and group D corresponding to an AUC value of 0.912 and between controls and two groups of metastatic cancers A and B with AUC values of 0.8052 and 0.9256, respectively. However, discrimination between metastatic hormone-sensitive (group A) and metastatic castration-resistant (group B) was low with an AUC of 0.512, revealing the much better performance of alpha satellite RNA (AUC 0.744) than PSA in discriminating two stages of metastatic prostate cancer. The correlation between alpha satellite RNA level and PSA level was also assessed in each group of patients (A–D) using Spearman’s rank correlation but no statistically significant correlation was found in any of the group (group A: r = −0.098, *p* = 0.7084; group B: r = 0.3978, *p* = 0.1602; group C: r = −0.2643, *p* = 0.1659; group D: r = −0.1056, *p* = 0.6968).

## 4. Discussion

Metastatic castration-resistant prostate cancer (mCRPC) is a clinical state in the trajectory of prostate cancer evolution characterised by disease progression despite patient castration of testosterone. Fundamentally, prostate cancer cells are exquisitely sensitive to testosterone suppression achieved by androgen-deprivation therapy via LHRH agonists. The hormone-sensitive state can last several years even in the presence of metastatic disease and is characterised by decreased or stable PSA and the resolution of metastatic lesions on imaging studies. However, castration resistance ultimately emerges as a consequence of the strong selective pressure of hormonal therapy exerted on prostate cancer cells [27]. In the spectrum of prostate cancer clinical course, mCRPC represents the final and incurable stage of disease with a survival rate of less than 3 years despite recent therapeutic breakthroughs. The key underlying mechanism of castration resistance is centred around the preserved activity of androgen receptor (AR) signalling, a crucial target of novel therapies for mCRPC. Different means of AR autonomy and therapeutic escape in mCRPC include AR protein overexpression, gene amplification, and/or AR mutations which all lead to the uninterrupted transduction of the AR signal and the intratumoural production of androgens [28,29,30].

At the genomics level, there are a number of differences between mCRPC and localised or hormone-sensitive metastatic prostate cancer. In addition to prevalent AR mutations, tumour samples from patients with mCRPC are enriched for mutations in *TP53*, DNA damage repair genes, *RB1* and *PTEN,* contributing to their loss-of-function and overall genomic instability [31,32]. Following that path, the discovery of targetable genetic defects in mCRPC opened a new avenue of targeted treatment, i.e., the use of PARP inhibitors for BRCA-mutated prostate cancer which is the most prevalent genomic event in mCRPC [33]. Recently, the co-existence of androgen-dependent and androgen-independent pathways was discovered in mCRPC, explaining the limited therapeutic efficacy of novel androgen pathway-targeted therapies in the general population of patients with mCRPC [34]. Finally, mCRPC is characterised by a large number of circulating tumour cells (CTCs) in the patient’s blood. Conversely, the presence of CTC in localised prostate cancer is considered to be an exceptional event. A less cohesive microenvironment of metastatic deposits in mCRPC facilitates the shedding of both individual tumour cells and cell-free DNA in the blood stream. Similar mechanisms may apply to the identification of alpha satellite RNA in the blood of patients with mCRPC.

The increased expression of pericentromeric satellite DNAs such as satellite II and alpha satellite DNA is characteristic of epithelial cancers including prostate cancer [7,35] as well as for hematopoietic malignancies [36]. Here, we observe significantly increased levels of alpha satellite RNA in the blood of patients with metastatic castration-resistant prostate cancer as well as an increase in patients with metastatic hormone-sensitive prostate cancer, while in the blood of patients with primary localised prostate cancer, no significant change relative to the controls was detected. One of the possible explanations for the increase in the alpha satellite RNA level in the blood of prostate cancer patients in metastatic stages of disease might be related to the transfer of alpha satellite RNA from prostate cancer cells to blood cells mediated by exosomes. The exosomes are a class of extracellular vesicles released by all cells, often detected in tumour microenvironments, which remove excess and/or unnecessary constituents from cells including harmful RNA and DNA [37,38]. Exosomes can transfer RNA or DNA which they contain to other cells, and in addition they can activate various signalling pathways in cells they fuse or interact with [38,39]. We propose that excess satellite RNA from prostate cancer can be transferred and delivered by exosomes to blood cells resulting in an increase in the total RNA level in blood cells. In addition, the interaction of exosomes with blood cells might activate some signalling pathways which could affect the heterochromatin structure and expression of satellite sequences located therein. In addition to the lysine-specific demethylase 2A (KDM2A) whose downregulation affects chromatin in prostate cancer [6], there are other (hetero)chromatin modifiers such as sirtuins, a family of NAD^+^-dependent deacetylases which coordinate cellular responses to different types of stress and are key players in the protection and maintenance of genomic integrity [40]. Of particular importance for the preservation of constitutive heterochromatin structure are the sirtuins SIRT1 and SIRT6 which maintain the epigenetic silencing of repetitive elements [41] by regulating the activity of the histone methyl-transferase SUV39H1 responsible for the spreading of the silencing mark H3K9me3 [40,42]. In addition, SIRT6 also deacetylates histone H3K18ac in pericentric heterochromatin and its depletion results in the overexpression of pericentromeric repeats [43]. It could be proposed that changes in the activation of KDM2A, sirtuins SIRT1 and SIRT6 or of some other (hetero)chromatin modifiers might arise not only in the prostate cancer cells [44] but possibly mediated by exosomes in the blood of prostate cancer patients at specific stages, affecting heterochromatin structure and the expression of repeats located therein. It is also possible that, as mentioned above, circulating tumour cells (CTCs) which are present in the blood of patients with metastatic prostate cancer [45] could precipitate with blood cells, contributing to the increased level of alpha satellite RNA, although due to the low number of CTCs relative to blood cells, this contribution is probably not significant.

As revealed by our results, the alpha satellite RNA level was able to discriminate metastatic hormone-sensitive from metastatic castration-resistant prostate cancer (groups A and B) as well metastatic castration-resistant cancer under treatment (B) from newly diagnosed localised prostate cancer before receiving a hormone or any local treatment (group D), from localised prostate cancer under treatment (group C) and from controls. On the other hand, PSA has high discrimination power for distinguishing controls from localised cancer before treatment (group D) as well as from metastatic cancers under treatment but cannot distinguish between metastatic hormone-sensitive and metastatic castration-resistant prostate cancers (groups A and B). Based on our investigation, the alpha satellite RNA level can complement PSA as a biomarker for monitoring the progression of metastatic prostate cancer and for the diagnosis of metastatic castration-resistant stage of disease. Considering the possible use of satellite RNA as a cancer biomarker, a circulating satellite RNA level in blood serum quantified by the sensitive method of tandem repeat amplification by nuclease protection (TRAP) combined with droplet digital PCR (ddPCR) enabled the discrimination of patients with pancreatic ductal carcinoma (PDAC) from healthy controls [46]. Increased levels of circulating human satellite II in the plasma of breast, gastric, lung and bile cancers as well as sarcoma and Hodgkin’s lymphoma was detected [47]. The present study reveals for the first time that not only serum or plasma-circulating satellite RNA but also alpha satellite RNA in blood cells could possibly serve as an indicator of a specific stage of cancer. In all these studies, the satellite RNA was used as a biomarker because its level is significantly increased in different cancers [7] and can be tested by quantitative real-time PCR or droplet digital PCR. Considering satellite DNA as a cancer biomarker, satellite copy number variation is characteristic for some cancers [48]; however, its detection is more complex and often requires a development of new assays [49] and technologies such as nanoplate-based digital PCR.

Further studies are necessary to explain the observed upregulation of intracellular alpha satellite RNA levels in the blood of prostate cancer patients at a specific metastatic stage and to disclose whether this phenomenon is specific to this pathological condition only.

## 5. Conclusions

The overexpression of satellite DNA is characteristic of many human cancers; however, it has not been investigated whether the satellite RNA level is changed in the whole blood of cancer patients. In this study, we analysed alpha satellite RNA level in the whole blood of prostate cancer patients at different stages of disease. The results reveal that the alpha satellite RNA level in the whole blood cells can discriminate castration-resistant metastatic prostate cancer from localised primary tumours and healthy controls as well as from metastatic hormone-sensitive prostate cancer with high accuracy. We discuss the possible mechanism which could result in the increased level of blood intracellular alpha satellite RNA at a specific metastatic stage of prostate cancer and propose alpha satellite RNA as a potential prostate cancer diagnostic blood biomarker.

## Figures and Tables

**Figure 1 genes-13-00383-f001:**
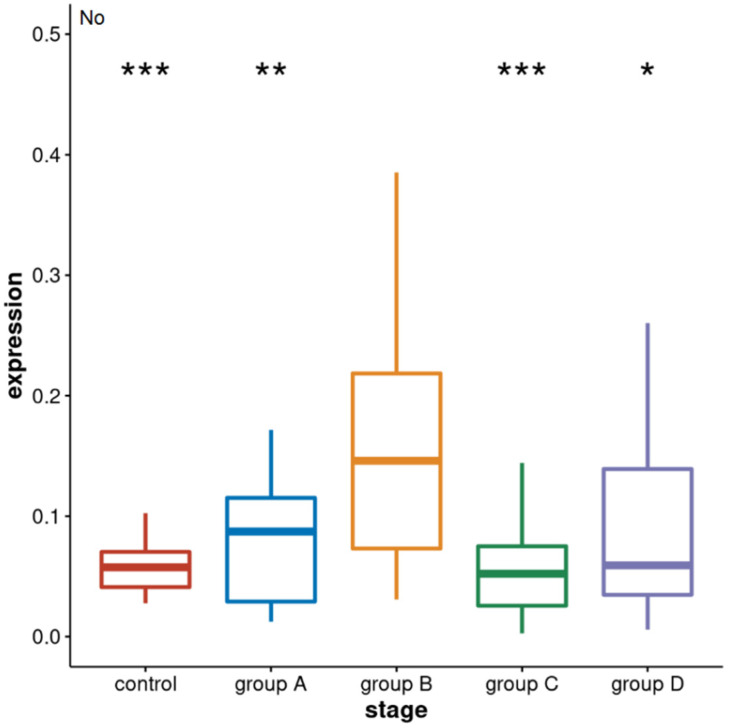
The level of intracellular alpha satellite RNA from the whole blood of control healthy individuals and of prostate cancer patients belonging to groups: A—metastatic hormone-sensitive on treatment; B—metastatic castration-resistant on treatment; C—localised hormone-sensitive on treatment; D—localised hormone-sensitive before any treatment. RNA level is obtained by RT-qPCR and the normalised average no value for each sample was used. Differences are analysed by 2-tailed Welch’s *t* test for groups A, B, and the control, and Mann–Whitney for group C and group D; median values are indicated and error bars represent standard deviations. Statistically significant differences of group B relative to groups A, C, D, and the controls, respectively, are indicated by stars (*** denotes *p* < 10^−^^3^, ** *p* < 10^−^^2^, * *p* < 0.05).

**Figure 2 genes-13-00383-f002:**
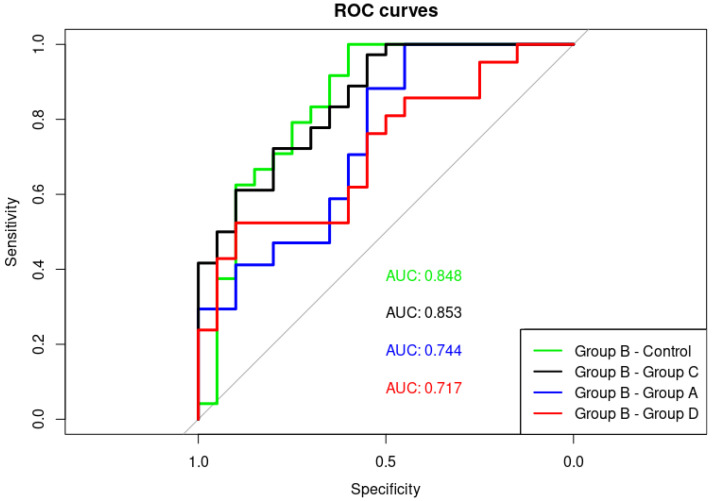
The diagnostic potential of alpha satellite RNA levels is determined by computing ROC curves and quantifying AUC values. The alpha RNA level shows the highest discriminatory power for distinguishing group B metastatic castration-resistant prostate cancer from: controls (AUC 0.848); group C (AUC 0.853); group A (AUC 0.744); and group D (AUC 0.717).

**Figure 3 genes-13-00383-f003:**
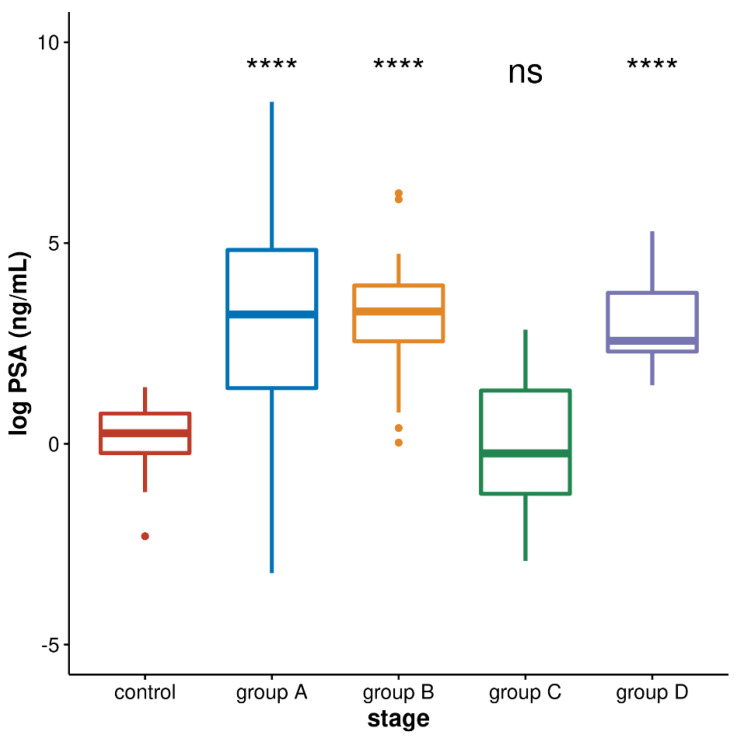
LogPSA values in the blood of control healthy individuals and of prostate cancer patients belonging to groups: A—metastatic hormone-sensitive on treatment; B—metastatic castration-resistant on treatment; C—localised hormone-sensitive on treatment; D—localised hormone-sensitive before any treatment. Differences between groups are analysed by Mann–Whitney and Kruskal–Wallis statistical tests, and median values are indicated and error bars represent standard deviations. Statistically significant differences of control relative to groups A, B. and D, respectively, are indicated by stars (**** denotes *p* < 10^−^^4^, ns means not significant).

**Table 1 genes-13-00383-t001:** Groups of individuals used in this study with the following characteristics: number of samples of each group (n); average age of patients (y); and the minimal and maximal age of individuals of each group.

Group	Characteristics
**Healthy controls**	
n	27
average age/y	39.4
age min–max/y	19–59
**Group A patients**	
n	19
average age/y	74.5
age min–max/y	62–85
**Group B patients**	
n	20
average age/y	67.4
age min–max/y	51–87
**Group C patients**	
n	34
average age/y	69.9
age min–max/y	57–83
**Group D patients**	
n	21
average age/y	71.9
age min–max/y	47–83

## Data Availability

Normalised no values representing the alpha satellite RNA expression level in patients belonging to four groups A–D and in healthy controls are presented in Appendix A of this paper.

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
