# Peer review of "Alpha Satellite RNA Levels Are Upregulated in the Blood of Patients with Metastatic Castration-Resistant Prostate Cancer"

_genes, 2022, doi:10.3390/genes13020383_

Round 1

Reviewer 1 Report

The authors provided an interesting paper concerning the role of alpha satellite RNA in prostate cancer.

The paper is nicely written and the results are clearly presented. My only concern is that the authors analysed the alpha satellite RNA and PSA levels and compared both of them as the markers for the castration-resistant tumour. However there are several papers concerning the problem and I don’t understand the need to evaluate the potency of PSA here again. However, if the authors would give some clarification why they chose to do so, it may be accepted.

Otherwise a nice manuscript.

Author Response

Reviewer 1

The authors provided an interesting paper concerning the role of alpha satellite RNA in prostate cancer.

  1. The paper is nicely written and the results are clearly presented. My only concern is that the authors analysed the alpha satellite RNA and PSA levels and compared both of them as the markers for the castration-resistant tumour. However there are several papers concerning the problem and I don’t understand the need to evaluate the potency of PSA here again. However, if the authors would give some clarification why they chose to do so, it may be accepted.

Otherwise a nice manuscript.

We agree with the reviewer that the potency of PSA as a marker for castration-resistant cancer was  previously investigated and in the introduction we commented that. We added PSA data to the manuscript to ilustrate and confirm previous studies and the data  could be removed, however Reviewer 2 asked for „additional PSA – alpha satellite RNA expression correlation analyses that would enhance the clinical relevance of the study“, and we decided to keep the data on PSA.

Reviewer 2 Report

Brief Summary: The aim of the study by Ljubic et al., was to assess the prognostic potential of alpha satellite RNA blood circulating levels for advanced or metastatic prostate cancer. The authors assessed the expression of alpha satellite RNA levels in the blood of patients with newly diagnosed localized, localized hormone-sensitive, metastatic hormone-sensitive or metastatic castration-resistant prostate cancer and in the blood of healthy individuals. They show that alpha satellite RNA levels are significantly higher in the blood from patients with metastatic castration-resistant prostate cancer (mCRPC) compared to healthy individuals, patients with localized or metastatic hormone-sensitive prostate cancer, suggesting that such analyses can be used to distinguish patients between disease stages. More specifically, the authors claim superiority of peripheral alpha satellite RNA expression analyses compared to the standard Prostate-specific Antigen (PSA) screening routinely preformed to determine the risk for prostate cancer. Their results show that in the same set of patients, PSA expression analyses alone failed to clearly distinguish patients with the most aggressive form of the disease, metastatic castration-resistant prostate cancer.  Overall, this is a well-designed study that presents novel and interesting data on the potential use of alpha satellite RNA to prognosticate prostate cancer progression. However, the manuscript would benefit from some clarifications and additional analyses to be considered for publication.

Strengths of the study: 

  • Analyses directly relevant to the clinical phenotypes of prostate cancer
  • Alpha satellite RNA expression as a potentially novel biomarker for advanced prostate cancer

Weaknesses of the study:

  • Lack of rationale for focus on alpha satellite RNA (why is it more important than alpha satellite DNA)
  • Lack of additional PSA – alpha satellite RNA expression correlation analyses that would enhance the clinical relevance of the study

Comments:

Abstract: This section is overall well-written but the authors should avoid claims of novelty like “for the first time”.

Introduction: This section presents background information on the satellite DNA and their potentila role in cancer development/progression. Comments:

  • This section lacks more information on prostate cancer and the need for new screening/biomarker approaches. The first paragraph of the discussion may be better utilized if modified for the introduction. [minor]
  • The authors present substantial background information on the alpha satellite DNA but do not mention its specific relevance or importance of alpha satellite RNA to prostate cancer or cancer in general. The rationale behind their study is not clear, as the benefit from analyzing alpha satellite RNA instead of DNA is not clear. [major]

Materials and Methods: The experimental procedures and study design are comprehensively explained

Results and Figures: The results are described adequately but the figures are lacking. Comments:

  • Figure 1 is confusing. The statistical analyses and comparisons are not clear and somewhat misleading in the figure. The authors should make clear between which groups there is a statistically significant difference either in the figure itself or at least in the figure legend. [major]
  • The information Figure 2 provides is unclear. What does AUC represent in this context and how is it relevant to the comparisons? Do these analyses provide any additional statistical significance information compared to Figure 1? The authors should clarify. [major]
  • In Figure 3, statistical significance analyses are missing but mentioned in the main text. [major]
  • The authors should definitely perform additional correlation analyses to determine whether alpha satellite RNA expression could enhance PSA screening accuracy or vise versa. This would significantly improve the impact of their study. [major]

Discussion and Conclusions: The authors present their findings but do not discuss them in adequate depth. Comments:

  • The study cohorts include healthy individuals of a much younger age than the prostate cancer patients (see Table 1). Could that difference in age explain for the difference in either PSA or alpha satellite RNA expression levels? The authors could explain why age-matched healthy individuals were not used for these analyses and discuss potential limitations of the study. [minor]
  • As mentioned above, the first two paragraphs of the discussion are background information on prostate cancer and more specifically mCRPC. Some of this information should be mentioned in the introduction to point out the need for new biomarkers and enhance the study rationale. [minor]
  • It is not clear why alpha satellite RNAs would be a better choice for prostate cancer screening than satellite DNAs. There is no discussion on the matter or any information on satellite RNAs and their relevance to cancer or/and prostate cancer. [major]
  • The authors suggest that exosomes may be carrying satellite RNAs in blood circulation. Why would exosomes preferably transport RNAs instead of DNAs? Is anything known on that aspect? This may provide a better rationale for studying satellite RNAs in favor of satellite DNAs. [minor]
  • The authors mention that alpha satellite RNA-containing circulating tumor cells (CTCs) may precipitate with blood cells, leading to their detection. Does this imply a limitation of the blood extraction process described in the Methods? It is supposedly designed to specifically extract blood cells. They should discuss and clarify this. [minor]

Author Response

Reviewer 2:

Comments:

  1.  Abstract: This section is overall well-written but the authors should avoid claims of novelty like “for the first time”.

It is deleted.

  1. Introduction: This section presents background information on the satellite DNA and their potentila role in cancer development/progression. Comments:

This section lacks more information on prostate cancer and the need for new screening/biomarker approaches. The first paragraph of the discussion may be better utilized if modified for the introduction. [minor]

The authors present substantial background information on the alpha satellite DNA but do not mention its specific relevance or importance of alpha satellite RNA to prostate cancer or cancer in general. The rationale behind their study is not clear, as the benefit from analyzing alpha satellite RNA instead of DNA is not clear. [major]

 In the Introduction we provide information about induced expression of satellite DNA in different cancers including prostate cancer and about the role of satellite transcripts in tumour progression (paragraph 2). The level of satellite RNA is significantly increased, up to 100x, in different cancers including prostate cancer (ref. 7) while the DNA level is generally not significantly changed. Since the level of satellite RNA is significantly increased in different cancers including prostate cancer we studied satellite RNA as a potential cancer biomarker. Considering DNA biomarkers, e.g. analysis of highly tandem repeat DNA -copy number variation is extremely hard if not impossible to do and should require a development of a specific nanoplate technology. We included in the Introduction additional paragraph 3 on PSA and the need for new biomarkers for prostate cancer (this paragraph was transferred from section Discussion).

  1. Materials and Methods: The experimental procedures and study design are comprehensively explained
  2. Results and Figures: The results are described adequately but the figures are lacking. Comments: Figure 1 is confusing. The statistical analyses and comparisons are not clear and somewhat misleading in the figure. The authors should make clear between which groups there is a statistically significant difference either in the figure itself or at least in the figure legend. [major]

 In the Figure 1 legend we added explanation for which pairs the statistically significant difference was presented.

  1. The information Figure 2 provides is unclear. What does AUC represent in this context and how is it relevant to the comparisons? Do these analyses provide any additional statistical significance information compared to Figure 1? The authors should clarify. [major]

We think that AUC values additionally clarify diagnostic potential of alpha satellite RNA and in the legend of Figure 2 we added explanation: „The diagnostic potential of alpha satellite RNA levels is determined by computing ROC curves and quantifying AUC values. The alpha RNA level shows the highest discriminatory power for distinguishing group B metastatic castration-resistant prostate cancer from: control (AUC 0.848), group C (AUC 0.853), group A (AUC 0.744) and group D (AUC 0.717). “

  1. In Figure 3, statistical significance analyses are missing but mentioned in the main text. [major]

In the Figure 3 legend we added explanation for which pairs the statistically significant difference was presented.

  1. The authors should definitely perform additional correlation analyses to determine whether alpha satellite RNA expression could enhance PSA screening accuracy or vise versa. This would significantly improve the impact of their study. [major]

We performed the correlation analysis  between  alpha satellite RNA level and PSA level in each group of patients (A-D) using Spearman's rank correlation but no statistically significant correlation was found in any of the group (group A: r=-0.098, P=0.7084; group B: r=0.3978, P=0.1602; group C: r=-0.2643, P=0.1659; group D: r=-0.1056, P=0.6968). This is written at the end of section Result: page 8. In the same paragraph we added report on ROC curve analysis of PSA levels which revealed discrimination between control and group D corresponding to AUC value of 0.912 and between control and two groups of metastatic cancers A and B with AUC values of 0.8052 and 0.9256, respectively. However, discrimination between metastatic hormone-sensitive and metastatic castration-resistant was low with AUC of 0.512, revealing much better performance of alpha satellite RNA  (AUC 0.744) than PSA  in discriminating two stages of metastatic prostate cancer.

We also added in Discussion, page 10, 2nd paragraph, the sentence” Based on our investigation alpha satellite RNA level can complement PSA as a biomarker for monitoring the progression of metastatic prostate cancer and for diagnosis of metastatic castration-resistant stage of disease.”

Discussion and Conclusions: The authors present their findings but do not discuss them in adequate depth. Comments:

  1.  The study cohorts include healthy individuals of a much younger age than the prostate cancer patients (see Table 1). Could that difference in age explain for the difference in either PSA or alpha satellite RNA expression levels? The authors could explain why age-matched healthy individuals were not used for these analyses and discuss potential limitations of the study. [minor]

50% of men older than 50 years have benign prostate hyperplasia (ref. 19) which can result in increased PSA level and to avoid this we used as healthy control group of younger men with average age of 39.4 years. This explanation is included in Materials and Methods, 2.2. Sample collection. We do not think that difference in age between controls and patients can explain difference in expression of satellite RNA. Namely, groups of patients differ significantly in satellite expression despite similarity in age. Also, patients groups C and D have levels of alpha satellite similar to the level in controls despite difference in age between control and the two patient groups.

  1. As mentioned above, the first two paragraphs of the discussion are background information on prostate cancer and more specifically mCRPC. Some of this information should be mentioned in the introduction to point out the need for new biomarkers and enhance the study rationale. [minor]

Paragraph dealing with necessity for new biomarker is transferred from Discussion to Introduction.

  1. It is not clear why alpha satellite RNAs would be a better choice for prostate cancer screening than satellite DNAs. There is no discussion on the matter or any information on satellite RNAs and their relevance to cancer or/and prostate cancer. [major]

The level of satellite RNA is significantly increased, up to 100x, in different cancers including prostate cancer while the DNA level is generally not significantly changed.  Increased levels of satellite RNA destabilize the replication fork and genome integrity and further promote tumour transformation. In Introduction, 2nd paragraph deals with increased expression of satellite DNA in cancer and role of satellite transcript.

  1. The authors suggest that exosomes may be carrying satellite RNAs in blood circulation. Why would exosomes preferably transport RNAs instead of DNAs? Is anything known on that aspect? This may provide a better rationale for studying satellite RNAs in favor of satellite DNAs. [minor]

Exosomes can carry both harmful DNA and RNA (ref. 37-39). However, since satellite RNA is present in huge amount in cancer cells it is proposed that the excess of it is carried by exosomes and the presence of satellite RNA in exosomes is experimentally confirmed (ref. 45).

  1.  The authors mention that alpha satellite RNA-containing circulating tumor cells (CTCs) may precipitate with blood cells, leading to their detection. Does this imply a limitation of the blood extraction process described in the Methods? It is supposedly designed to specifically extract blood cells. They should discuss and clarify this. [minor]

The method was designed to trigger the complete lysis of all cells in the blood sample and precipitated cellular debris (precipitated at 5000 g) are used for RNA isolation. Therefore, it is probable that circulating tumour cells are also lysed and that their debris co-precipitated with those of blood cells. However, the number of circulating tumour cells is very low relative to the number of blood cells and we think that their contribution to alpha satellite expression is low but needs to be mentioned in Discussion. We added to the sentence in the Discussion 1st paragraph, page 10 “due to low number of CTC relative to blood cells this contribution is probably not significant”.

Round 2

Reviewer 2 Report

The authors addressed the majority of the reviewer’s concerns, and the manuscript has improved. Although there is more information on prostate cancer and the need for new screening/biomarker approaches, focusing on PSA screening, the authors still fail to address the importance or potential superiority of satellite RNA over DNA for biomarker-based screening approaches. They provide a rather good explanation in their response to reviewer’s but do not include that in the main text of the revised manuscript. [minor]

Author Response

The authors addressed the majority of the reviewer’s concerns, and the manuscript has improved. Although there is more information on prostate cancer and the need for new screening/biomarker approaches, focusing on PSA screening, the authors still fail to address the importance or potential superiority of satellite RNA over DNA for biomarker-based screening approaches. They provide a rather good explanation in their response to reviewer’s but do not include that in the main text of the revised manuscript. [minor]

We added in the Discussion comment on satellite RNA and satellite DNA as cancer biomarkers (page 10, end of the penultimate paragraph) “In all these studies the satellite RNA was used as a biomarker because its level is significantly increased in different cancers [7] and can be tested by quantitative real-time PCR or droplet digital PCR. Considering satellite DNA as a cancer biomarker, satellite copy number variation is characteristic for some cancers [48], however its detection is more complex and often requires a development of new assays [49] and technologies such as nanoplate-based digital PCR.